# The Use of UAV Mounted Sensors for Precise Detection of Bark Beetle Infestation

**Tomáš Klouček** [1] , **Jan Komárek** [1,*] , **Peter Surový** [2] , **Karel Hrach** [1] , **Přemysl Janata** [3] and **Bedřich Vašíček** [3]

[1]  Department of Applied Geoinformatics and Spatial Planning, Faculty of Environmental Sciences, Czech University of Life Sciences Prague, Kamýcká 129, Praha - Suchdol, 165 00 Prague, Czech Republic

[2]  Department of Forest Management, Faculty of Forestry and Wood Sciences, Czech University of Life Sciences Prague, Kamýcká 129, Praha - Suchdol, 165 00 Prague, Czech Republic

[3]  The Krkonose Mountains National Park Administration, Dobrovského 3, 543 01 Vrchlabí, Czech Republic

*  Correspondence: komarekjan@fzp.czu.cz; Tel.: +420-224-383-837

**Abstract:** The bark beetle *(Ips typographus)* disturbance represents serious environmental and economic issue and presents a major challenge for forest management. A timely detection of bark beetle infestation is therefore necessary to reduce losses. Besides wood production, a bark beetle outbreak affects the forest ecosystem in many other ways including the water cycle, nutrient cycle, or carbon fixation. On that account, (not just) European temperate coniferous forests may become endangered ecosystems. Our study was performed in the unmanaged zone of the Krkonoše Mountains National Park in the northern part of the Czech Republic where the natural spreading of bark beetle is slow and, therefore, allow us to continuously monitor the infested trees that are, in contrast to managed forests, not being removed. The aim of this work is to evaluate possibilities of unmanned aerial vehicle (UAV)-mounted low-cost RGB and modified near-infrared sensors for detection of different stages of infested trees at the individual level, using a retrospective time series for recognition of still green but already infested trees (so-called green attack). A mosaic was created from the UAV imagery, radiometrically calibrated for surface reflectance, and five vegetation indices were calculated; the reference data about the stage of bark beetle infestation was obtained through a combination of field survey and visual interpretation of an orthomosaic. The differences of vegetation indices between infested and healthy trees over four time points were statistically evaluated and classified using the Maximum Likelihood classifier. Achieved results confirm our assumptions that it is possible to use a low-cost UAV-based sensor for detection of various stages of bark beetle infestation across seasons; with increasing time after infection, distinguishing infested trees from healthy ones grows easier. The best performance was achieved by the Greenness Index with overall accuracy of 78%–96% across the time periods. The performance of the indices based on near-infrared band was lower.

**Keywords:** bark beetle detection; spectral change; UAVs; green attack; forest infestation; near infrared (NIR); visible spectrum

## 1. Introduction

The current climate change causes serious difficulties for forests and keeping track of natural hazards such as pest outbreaks represents therefore a major challenge for forest management and for the future of forest ecosystems [1–4]. Over the last decades, spruce forests (not only) in the Central Europe have been affected by ever increasing bark beetle activity. Thus, coniferous forests, suffering besides pest attacks also from frequent windstorms and droughts, may join the ranks of endangered ecosystems in the foreseeable future. The probability of bark beetle attack increases after long periods

of drought, which debilitates the natural defences of the trees. Above-average temperatures and below-average precipitation in the last four years weakened the forest ecosystems and set the conditions for biotic infestation.

European spruce bark beetle (*Ips typographus*) is a secondary pest that infests primarily recently-harvested wood or weakened trees of spruce [5,6]; however, where overpopulation occurs, they can also infest healthy trees. An overabundance of bark beetle affects not only the production of wood matter; it also affects other forest functions [7] such as water retention or carbon sequestration, nutrients storage [8], or biodiversity [9]. Besides such environmental concerns, there are of course also economic impacts. A bark beetle outbreak causes a significant drop in the value of wood and, even more importantly, imposes significant costs associated with its consequences and recovery [10].

As prevention is the most effective defence against bark beetle, it is necessary to focus on the deceleration of its spread; however, the spatial and temporal dynamics of the pest's disturbances are not fully understood yet [11,12]. For the measures to be successful, the earliest possible detection of infested trees is needed [13]. An infested tree starts to evince visual changes as well as changes in spectral characteristics as soon as a few weeks after infestation [14]. Such spectral changes may be recorded using remote sensing (RS) techniques, i.e., satellites, aeroplanes, or unmanned aerial vehicles (UAVs). As the satellites are used to map spatiotemporal dynamics of the infestation spread at large extents, drones may be used for detection of the infested trees at a very detailed scale; see [11] for review. A comprehensive review of assessing the health status of trees using remote sensing is provided by [15,16].

Invasive approaches (felling the infested trees, peeling the bark off, application of chemicals, etc.) for deceleration of bark beetle activity are not applicable in protected areas such as non-intervention zones of national parks where only autoregulatory conservation management (succession) is possible. These areas are a valuable source of information about the bark beetle life cycle since no human-driven management interferes with spreading of the beetle. Moreover, these zones are usually situated in locations that are difficult to access. Thanks to detectable changes in spectral characteristics between healthy and infested trees [4,14], the use of RS techniques seems to be a very promising solution [2,17].

The satellite-borne multispectral imagery [11–13], hyperspectral imagery [4,5,18,19], airborne LiDAR [7,20], or a combination of those have been applied to study bark beetle across large extents. On the other hand, the close-range RS techniques (e.g., drones) may be used for precise detection of infested trees at a very detailed scale [4,21–24]. As the UAVs (or drones) have gained in popularity and use, their prices have come down and the availability of user-friendly software has increased. Their usability for early detection of infested trees is however still limited and the selection of appropriate time of UAV data acquisition for early detection still missing. Despite that, UAVs are advantageous due to their (a) spatial resolution, which offers a solution for local scale analysis at the level of individual trees [25]; and (b) temporal resolution where rapid deployment is crucial [26]. Miniaturized UAV-specific sensors thus represent a state-of-the-art solution for many recent environmental applications [22,27] and deriving forestry parameters [28,29]. Despite this, only few studies have focused on bark beetle detection using UAVs [30,31]. For example, Näsi et al. [4] distinguished between healthy, infested, and dead trees with an overall accuracy of 76% and Näsi et al. [19] with 81% overall accuracy using hyperspectral UAV-borne images in Finland; however, they did not distinguish between healthy trees and those in an early stage of infestation. The potential of the application of vegetation indices for detection of trees infested with bark beetle in different environments was previously reported by Näsi et al. [19] in Finland; Minařík et al. [23] in the Czech Republic; Stoyanova et al. [21] in Bulgaria, and Safonova et al. [32] in Russia. Despite the fact that the significance of NIR band was proved by [14,21–23], the detection of the early stage of the bark beetle attack, even before the visual signs are easily recognizable by the human eye (so-called "green attack"), is still challenging, especially using a low-cost camera.

The bark beetle infestation has several phases, which are annually repeated [5,6]. The aim of this study is to highlight the possibility of using low-cost and customized UAV sensors for detection

of various stages of bark beetle attack at the level of individual trees. We hypothesize that (a) using broad UAV-based bands, it is possible to detect spectral differences between healthy and infected trees at various stages, and (b) the near-infrared band plays a more important role in the detection of trees infested by bark beetle than RGB bands, even when originating from a low-cost, customized RGB camera. At the first glance, the study represents a typical environmental application of remote sensing; in a more detailed view, however, it is unique due to (a) the use of low-cost sensors placed on rotary-wing UAV for bark beetle detection; (b) coupling a customized NIR sensor with a consumer grade RGB camera; (c) distinguishing between healthy and infested trees in the green attack stage; (d) a detailed spatial analysis at the level of individual trees (sub-centimetres image resolution); and (e) location of the study area in the non-intervention zone, which allowed us to comprehensively describe the forest changes throughout one bark beetle generation.

## 2. Materials and Methods

### 2.1. Study Site

The study area with the elevation range of 940–1050 m above MSL is located in the eastern part of the biggest national park (Krkonoše Mountains National Park) in the north of the Czech Republic (Figure 1). A state border divides the park into Czech and Polish parts with the combined area of 425 km². It was founded in 1963 and has been listed as a UNESCO Biosphere Reserve site in 1992. The study area encompasses 10.7 ha and is covered mainly by spruce forest (*Picea abies)*, supplemented by mountain-ash (*Sorbus aucuparia)*, common beech (*Fagus sylvatica),* and silver fir (*Abies alba).* The mean annual temperature is less than 5°C and the mean annual precipitation exceeds 1400 mm; the duration of vegetation season is approx. a hundred days. The study site, which is included in the first level of the protected areas (the most strictly protected), is a part of a non-intervention zone left strictly to ecological succession—no human-managed interventions are permitted. The study area had a high probability of bark beetle activity in the year of data acquisition due to its activity in the past.

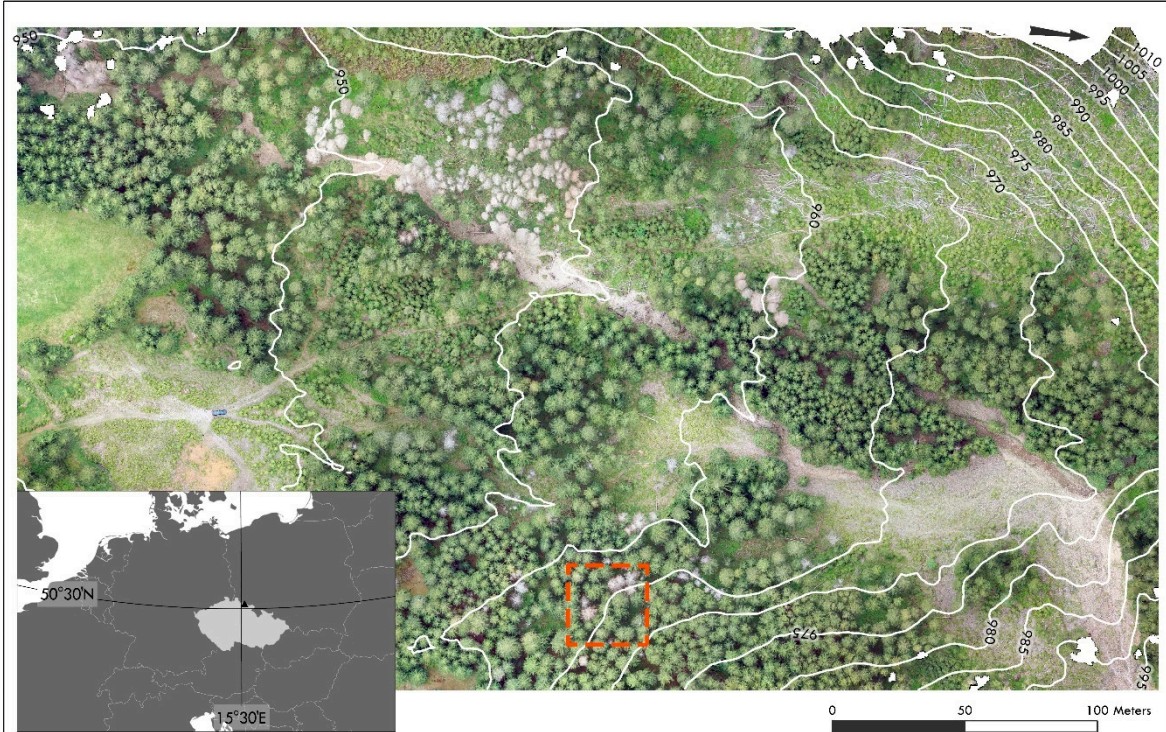

**Figure 1.** The location of the study area (50°30′N 15°30′E) in the central Europe and the extent of the study site; the red square denotes the borders of Figure 2. The background image represents the true color composite for the first sensing period (15 June).

The bark beetle attack in this area starts usually later than in other parts of the Czech Republic due to the high altitude and snowy conditions. It is hard to specify the exact date but typically, it is the end of May (last week of May). The records by local foresters indicate that in 2017, swarming took place on 1–10 June (sinkholes, wedding chambers and eggs were found in the traps). In the next period (15–20 June), the larval stage was found in most of the traps. At the turn of July and August, development progressed to the pupal stage and young beetles started to emerge in the middle of August (after 15 August). Visual infestation stages corresponded with this statement. Hence, the trees infested in the orthomosaics from the first (15 June) and second (1 August) periods are in the green stage, in the third (30 August) mainly in the yellow stage and in the fourth (1 October) mainly in the red stage.

## 2.2. Acquisition and Processing of UAV Imagery

Field surveys were performed four times during the 2017 season: (a) At the beginning of the outbreak (on 15 June), (b, c) during the outbreak on 1 August (b) and on 30 August (c), and (d) after the outbreak (on 1 October). The UAV flights in June and in October were performed in overcast conditions while the flights in August in sunny conditions (see Figure A2). Where the images were acquired under overcast conditions, the cloud cover of the sky was consistent during all UAV missions and in sunny periods, almost no clouds that could have caused problematic shadows were present in the sky. Each flight lasted for 15 min so we were able to wait for the appropriate time of image acquisition—the individual sets of images acquired on the same days were acquired under comparable conditions. All flights were performed at almost the same time (11:00–13:00). Data was acquired during a low altitude aerial survey using a rotary-wing UAV Zefyros Oktos XL (MikroKopter, Moormerland, Germany). UAV was equipped with (a) a casual camera Sony Alpha A7 and (b) a customized sensor Lumix TZ7. Sony Alpha A7, which is a CMOS-based full-frame 24 MPix camera, was used for acquiring visible imagery to make a true-color mosaic (RGB); the camera was equipped with a 21 mm focal length lens providing low geometrical distortions. Lumix TZ7, a CCD-based 10 MPix camera, was customized to capture near-infrared spectrum as well and thus to obtain a color infrared mosaic (CIR). The NIR filter was removed by a member of the team of authors and a 760 nm band filter was mounted in front of the lens to filter out the visible spectrum. Both cameras were mounted on the same UAV.

Predefined flight plan at 90 m above ground level with regular 80% side and frontal overlaps was uploaded to the UAV control/autopilot unit. Seven Ground Control Points distributed regularly throughout the area were surveyed using GNSS aperture Topcon Hiper HR (the mean RMSE in XYZ axis was 0.03 m) to build a strong geometry of the photogrammetric model. In total, 673 camera stations were taken above the study site per flight.

UAV-acquired images were processed using image-matching software PhotoScan version 1.3.4 (Agisoft LLC, Saint Petersburg, Russia) without initial camera calibration parameters, see Figure A1 for image residuals. More than 600 thousand key points were identified during the image alignment process and subsequently densified. Orthorectified mosaics were built with a ground sampling distance of 2.3 cm; all four flight campaigns were processed using the same processing parameters. The output from the image-matching processing was a combination of RGB and CIR orthomosaics for each field campaign. A Digital Surface Model (DSM) with the density of 125 points per square meter was created. Subsequently, a Digital Terrain Model (DTM) was built and subtracted from the DSM to acquire a Canopy Height Model (CHM) representing the vegetation height.

## 2.3. Image Analysis

Four composite RGB/CIR orthomosaics were created and radiometrically calibrated to relative surface reflectance using the Flat Field Correction feature in ENVI software version 5.5 (Harris Geospatial Solutions Inc., Broomfield, CO, USA). The correction was necessary to adjust for the different radiometric resolutions of the used cameras and to remove the differences in weather and seasonal conditions between UAV sensing periods. Flat Field Correction works by dividing the

reflectance values of individual pixels in the individual channels by a mean value from a user-defined reference area containing spectrally flat material. The only suitable type of such material in the study area were boulders in the dry river basin.

Shadows were masked off from the composite orthomosaics using the near-infrared band. The shadows masks were created using thresholding. The threshold value was set manually (trial-and-error method) for each of the four composites. To achieve precise detection, individual treetops were automatically detected using CHM and local maxima filtering [28,29]. This widely used approach was based on a proprietary script using image smoothing by the low pass 3-by-3 filter, local maxima finding with a 2 m radius, and template matching [28] in ArcGIS software version 10.6 (ESRI, Redlands, CA, USA). Detected treetops were visually inspected in the orthomosaic and manually corrected in a few cases where multiple local maxima were found on a single tree. As the bark beetle prefers spruce trees older than 60 years that provide a plentiful source of phloem [12], trees lower than 15 m were ignored during the detection process.

Each treetop was classified according to its health status (dead, healthy, or infested) for each sensing period. This classification was based on the visual image interpretation of the differences between the state of the tree in the first and the last orthomosaics and verified by a field survey. As the spectral characteristics of dead trees significantly differed from both healthy and infested trees [4], dead trees were not included into subsequent statistical analysis and classification process. From processed orthomosaics, selected vegetation indices were calculated [33,34], see Table 1. The relevance of indices was evaluated based on visual differences in spectral curves (bands) of infested and healthy trees (see Figure 2). Around every treetop, a 0.5 m buffer was created to eliminate shadows of surrounding trees; moreover, an assumption was that the first symptoms were visible near the treetop of the infested tree. In each buffer, mean vegetation indices were calculated using Zonal Statistics tool in ArcGIS for further statistical analysis.

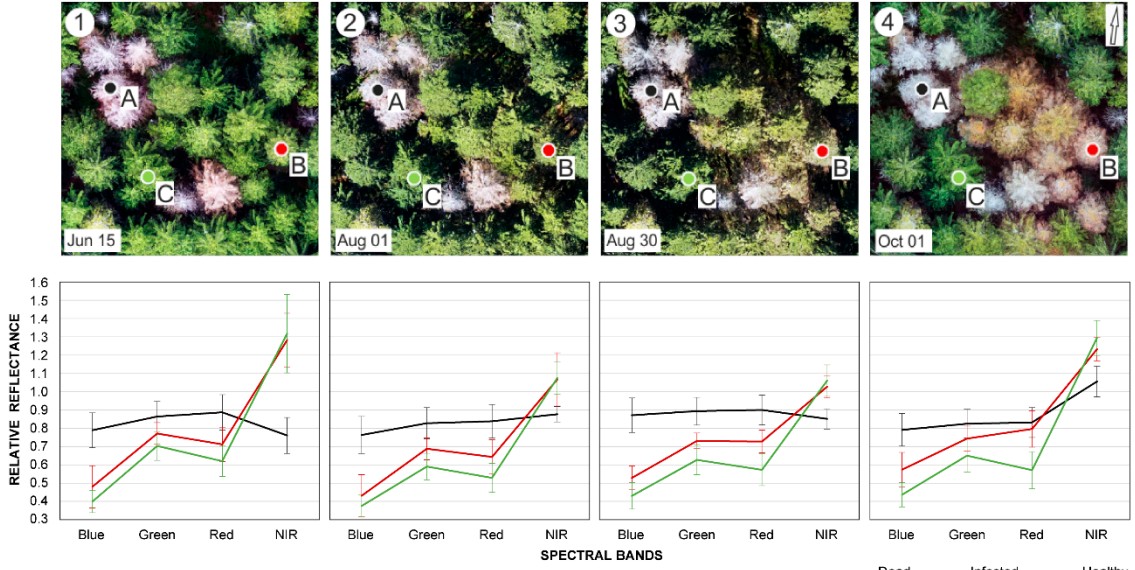

**Figure 2.** Spectral curves of the time series captured in different stages of the bark beetle infestation, showing dead (grey, A), infested (red, B), and healthy (green, C) trees throughout the season. The graphs represent mean relative reflectance values calculated from all infested (red), dead (black), and healthy (green) trees higher than 15 m in the study area at the individual dates.

**Table 1.** Calculated broadband ratio indices with formulas.

| Indices | Formula | Reference |
|---|---|---|
| Simple Ratio | $SR = \frac{NIR}{RED}$ | [35] |
| Greenness Index | $GI = \frac{GREEN}{RED}$ | [36] |
| Green Ratio Vegetation Index | $GRVI = \frac{NIR}{GREEN}$ | [37] |
| Normalized Difference Vegetation Index | $NDVI = \frac{NIR-RED}{NIR+RED}$ | [38,39] |
| Green Normalized Difference Vegetation Index | $GNDVI = \frac{NIR-GREEN}{NIR+GREEN}$ | [40] |

## 2.4. Statistical Analysis

The main aim of the statistical analysis was to identify possibilities of distinguishing the healthy trees from the infested ones throughout the sensing periods. The selected infested trees (24 trees in total) were compared with a random sample selected from the set of healthy trees (3226 trees in total). The difference was tested using Mann–Whitney test within each observational period separately. The size of the difference and its change in time were assessed using the Relative Treatment Effect (RTE) values. STATISTICA software version 13.4 (TIBCO Software Inc., Palo Alto, CA, USA) was used for Mann–Whitney tests, R environment version 3.5.1 (R Core Team, Vienna, Austria) for RTE calculations using nparLD procedure as described in [41].

The healthy (H-sample) and infested (I-sample) trees were compared by statistical analysis. Infested trees were included in the I-sample (22 trees in total, two samples were excluded due to a low number of pixels in the 0.5 m buffer) while H-sample was a random sample selected from the set of healthy trees, based on the following criteria: the minimum tree-height 15 m, the maximum distance to the nearest infested tree 50 m, the minimum number of pixels in the treetop buffer 500 (2138 healthy trees met the sampling criteria). For a graphical comparison of selected samples, see Figure 3.

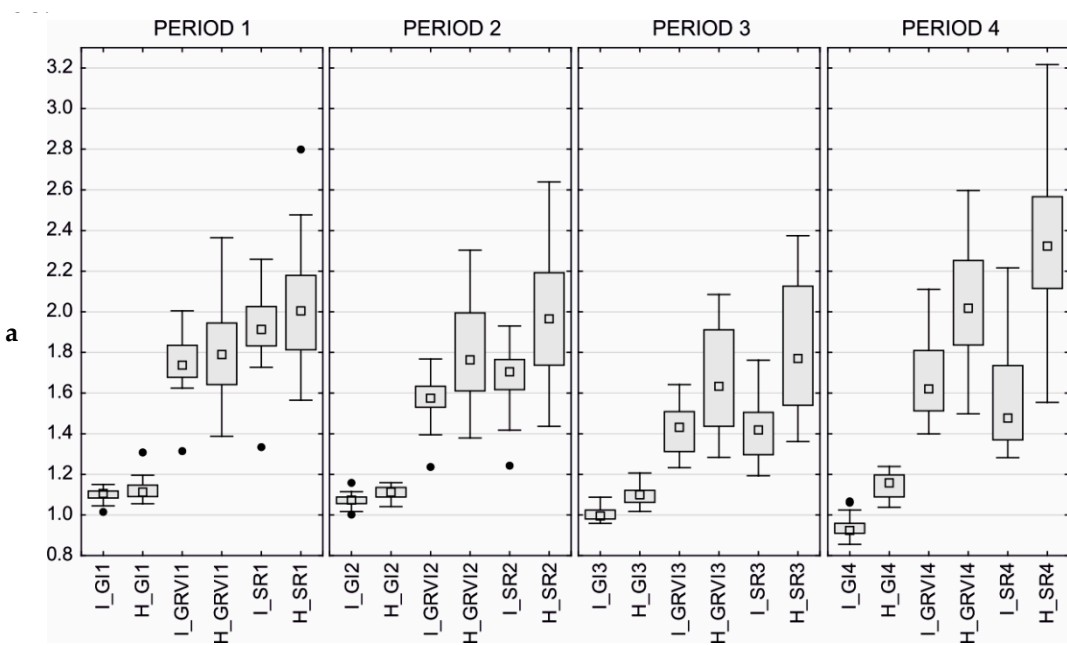

**Figure 3.** *Cont.*

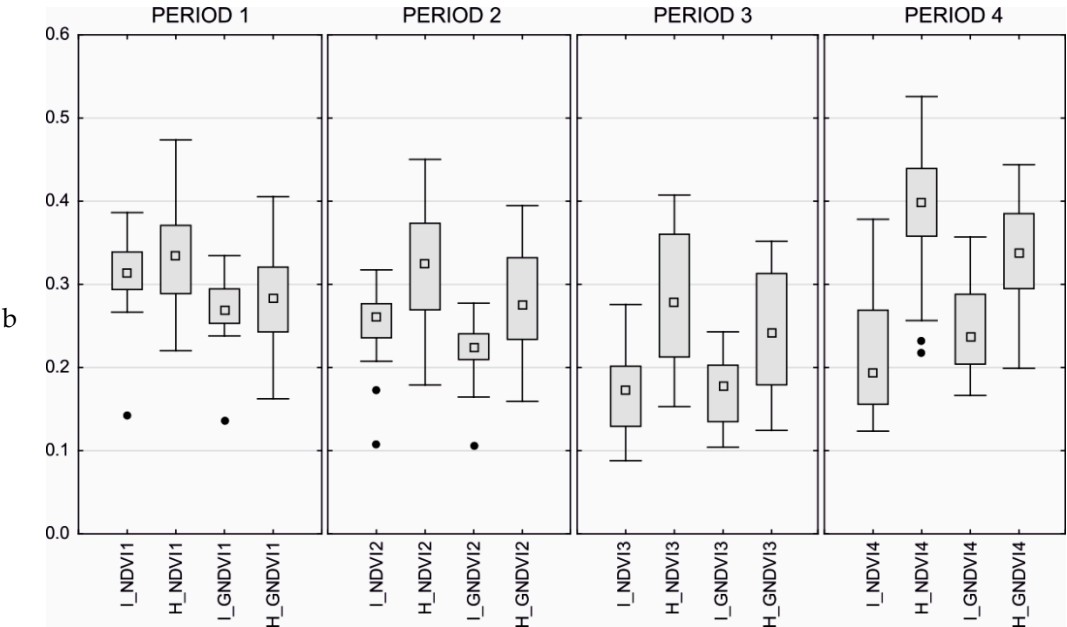

**Figure 3. (a)** Box-plots displaying quartile characteristics (median, Q25, Q75) for SR, GI, and GRVI, respectively (always in the order I-sample—infested, H-sample—healthy), in the four sensing periods. Black dots represent outliers; **(b)** Box-plots displaying quartile characteristics (median, Q25, Q75) for NDVI and GNDVI, respectively (always in the order I-sample—infested, H-sample—healthy), in the four sensing periods. Black dots represent outliers.

### 2.5. Image Classification

Statistical results were compared to the results of image classification using Maximum Likelihood Classifier (MLC). Samples of 40 healthy and 10 infested trees were randomly selected for training as well as for validation. Classification was performed in ArcGIS using the following inputs: Greenness Index (GI), Simple Ratio (SR), Green Ratio Vegetation Index (GRVI), Normalized Difference Vegetation Index (NDVI), and Green Normalized Difference Vegetation Index (GNDVI), see Table 1 for details.

## 3. Results

### 3.1. Spectral Comparison

The spectral response of identified trees varied across the dates of acquisition (Figure 2). The imagery analysis was used to determine spectral profiles of the healthy, dead, and infested trees throughout the season. The spectral differences between healthy and infested trees were distinguishable both in the visible and near-infrared bands and kept increasing with later dates of acquisition. The spectral differences between the infested and healthy trees were higher in the visible bands than in the near-infrared bands.

The spectral response of dead trees is almost constant across different stages, changing towards the end, probably due to changes in spectral reflectance caused by falling off of needles and fine branches. The mean reflectance values are therefore very similar in visible bands and are different in the NIR band (see Figure 2). The response of both infested and healthy trees changed over time. Nevertheless, in early summer (15 June), at the beginning of infestation, the shapes of the spectral curves were quite similar. There was no significant difference between healthy and infested trees (Figure 2). In summer (1 August and 30 August), when the bark beetle attack progressed, the humidity was lower and temperature higher, the response of healthy and infested trees started to differ and a spectral variation was apparent, mainly in Red bands. In autumn (1 October), the difference in the Red band between healthy and infested trees is clearly visible. In addition, the relative spectral reflectance of infested trees was getting closer to that of dead trees (see the graphs in Figure 2).

### 3.2. Statistical Evaluation

Due to the lack of normality, a non-parametric test procedure (Mann–Whitney test) was used to detect significant differences between samples (see Hypothesis a), separately for each observation time (Period 1–Period 4). Resulting p-values indicate that the difference between healthy and infested trees was significant even at the early stage, namely since the second period of observation (Table 2). An expected effect of enhancing the detection capabilities by adding the NIR band (see Hypothesis b) was not observed because all the indices showed a comparable ability in detecting the differences between healthy and infested trees. The NIR however contributed to the results by yielding the best shadow mask.

**Table 2.** Comparison of median values in the H-sample/I-sample (p-values of Mann–Whitney test in brackets). Note: The resulting p-values for GRVI and GNDVI, as well as for SR and NDVI, were identical in these non-parametric test procedures due to the definition of the indices (see Table 1).

|  | Period 1 | Period 2 | Period 3 | Period 4 |
|---|---|---|---|---|
| **SR** | 2.007/1.911 (0.342) | 1.966/1.706 (0.001) | 1.771/1.420 (<0.001) | 2.326/1.478 (<0.001) |
| **GI** | 1.112/1.106 (0.245) | 1.115/1.075 (0.001) | 1.097/0.995 (<0.001) | 1.156/0.921 (<0.001) |
| **GRVI** | 1.793/1.735 (0.581) | 1.762/1.576 (0.002) | 1.636/1.431 (0.001) | 2.020/1.620 (<0.001) |
| **NDVI** | 0.335/0.313 (0.342) | 0.326/0.261 (0.001) | 0.278/0.173 (<0.001) | 0.399/0.193 (<0.001) |
| **GNDVI** | 0.284/0.269 (0.581) | 0.276/0.223 (0.002) | 0.241/0.177 (0.001) | 0.338/0.237 (<0.001) |

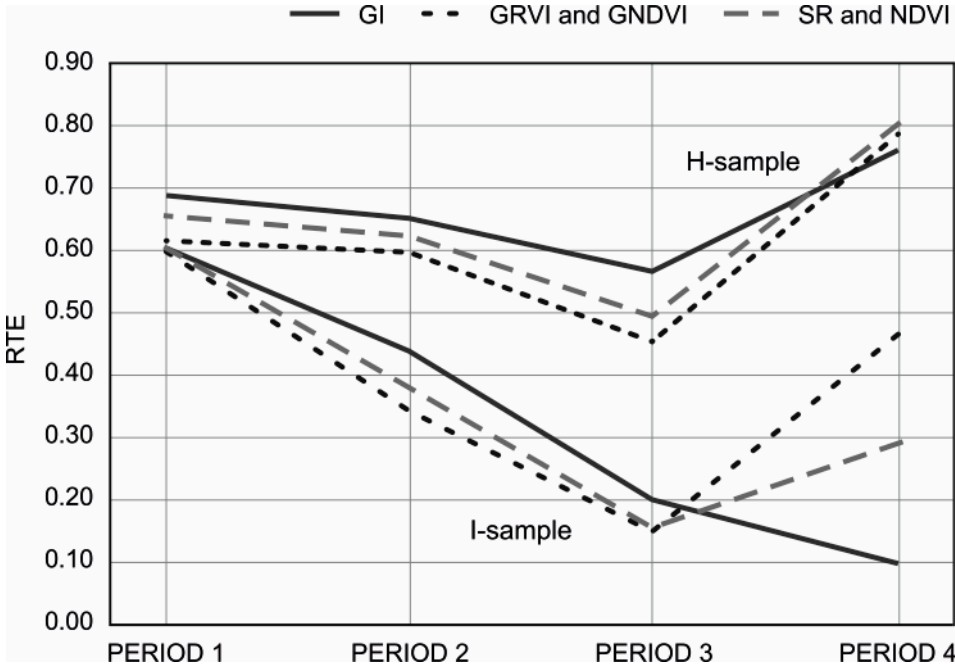

**Figure 4.** Relative Treatment Effect (RTE) values (GI: solid curves, GRVI and GNDVI: dotted curves, SR and NDVI: dashed curves; H-sample—upper curves, I-sample—lower curves).

The sampling of healthy trees and the testing procedure were repeated 100-times to assess the stability of the result. In 56 cases (56%), GI differed significantly between healthy and infested trees as soon as in the first period of observation; SR as well as NDVI recorded the change at that stage in 39 cases (39%); and GRVI and GNDVI in 21 cases (21%). The increasing difference between the H-sample and I-sample over time is obvious from Table 2 as a decrease in p-values from Period

1 to Period 4. As an illustration of this trend, RTE values were calculated, as shown in Figure 4. The differences between H-samples (upper curves) and I-samples (lower curves) increased with later periods. For instance, the RTE value for GI in the first period reached 68.5% for H-sample and 60.4% for I-sample; in the second period, it reached 65.2% for H-sample but only 44.0% for I-sample. The last value, for example, can be interpreted as follows: with a probability of 44%, any GI value measured among the infested trees in the second period tends to be smaller than any GI value from the whole I-sample dataset.

### 3.3. Image Classification

The Greenness Index yielded the best image MLC classification results for the first period (at the beginning of the bark beetle outbreaks). Moreover, the classification accuracy increased with later time of image acquisition for all indices (e.g., GI 78%–96%, NDVI 70%–94% and GNDVI 60%–92%). In addition, classification results slightly overestimated the number of the infested trees; for practical forest application, however, this type of mistake is less harmful than the opposite. The bark beetle infestation may be captured in the early stages due to omission-free results. In addition, the small number of healthy trees misclassified as infested represent unhealthy trees, which could be potentially attacked by bark beetle in the next generation. Therefore, their removal could be also beneficial.

**Table 3.** Image classification accuracy expressed on a sample of 10 infested:40 healthy trees using maximum likelihood classification where H = number of healthy trees, I = number of infested trees, $\Sigma$ = sum, $P_A$ = producer's accuracy, $U_A$ = user's accuracy, **bold values** = overall accuracy. The graphical interpretation of the classification approaches may be found in Figure A2 in Appendix C.

| Time | | 15 June 2017 | | | | 1 August 2017 | | | | 30 August 2017 | | | | 1 October 2017 | | | |
|---|---|---|---|---|---|---|---|---|---|---|---|---|---|---|---|---|---|
| | | H | I | $\Sigma$ | $U_A$ | H | I | $\Sigma$ | $U_A$ | H | I | $\Sigma$ | $U_A$ | H | I | $\Sigma$ | $U_A$ |
| **GI** | H | 30 | 1 | 31 | 0.97 | 33 | 1 | 34 | 0.97 | 38 | 2 | 40 | 0.95 | 39 | 1 | 40 | 0.98 |
| | I | 10 | 9 | 19 | 0.47 | 7 | 9 | 16 | 0.56 | 2 | 8 | 10 | 0.80 | 1 | 9 | 10 | 0.90 |
| | $\Sigma$ | 40 | 10 | 50 | | 40 | 10 | 50 | | 40 | 10 | 50 | | 40 | 10 | 50 | |
| | $P_A$ | 0.75 | 0.90 | | **0.78** | 0.83 | 0.90 | | **0.84** | 0.95 | 0.80 | | **0.92** | 0.98 | 0.90 | | **0.96** |
| **NDVI** | H | 28 | 3 | 31 | 0.90 | 32 | 1 | 33 | 0.97 | 36 | 3 | 39 | 0.92 | 39 | 2 | 41 | 0.95 |
| | I | 12 | 7 | 19 | 0.37 | 8 | 9 | 17 | 0.53 | 4 | 7 | 11 | 0.64 | 1 | 8 | 9 | 0.89 |
| | $\Sigma$ | 40 | 10 | 50 | | 40 | 10 | 50 | | 40 | 10 | 50 | | 40 | 10 | 50 | |
| | $P_A$ | 0.70 | 0.70 | | **0.70** | 0.80 | 0.90 | | **0.82** | 0.90 | 0.70 | | **0.86** | 0.98 | 0.80 | | **0.94** |
| **SR** | H | 24 | 2 | 26 | 0.92 | 30 | 1 | 33 | 0.97 | 35 | 3 | 38 | 0.92 | 38 | 2 | 40 | 0.95 |
| | I | 16 | 8 | 24 | 0.33 | 10 | 9 | 17 | 0.47 | 5 | 7 | 12 | 0.58 | 2 | 8 | 10 | 0.80 |
| | $\Sigma$ | 40 | 10 | 50 | | 40 | 10 | 50 | | 40 | 10 | 50 | | 40 | 10 | 50 | |
| | $P_A$ | 0.60 | 0.80 | | **0.64** | 0.75 | 0.90 | | **0.78** | 0.88 | 0.70 | | **0.84** | 0.95 | 0.80 | | **0.92** |
| **GNDVI** | H | 23 | 3 | 26 | 0.89 | 30 | 1 | 31 | 0.97 | 35 | 4 | 39 | 0.90 | 38 | 2 | 40 | 0.95 |
| | I | 17 | 7 | 24 | 0.29 | 10 | 9 | 19 | 0.47 | 5 | 6 | 11 | 0.55 | 2 | 8 | 10 | 0.80 |
| | $\Sigma$ | 40 | 10 | 50 | | 40 | 10 | 50 | | 40 | 10 | 50 | | 40 | 10 | 50 | |
| | $P_A$ | 0.58 | 0.70 | | **0.60** | 0.75 | 0.90 | | **0.78** | 0.88 | 0.60 | | **0.82** | 0.95 | 0.80 | | **0.92** |
| **GRVI** | H | 22 | 2 | 24 | 0.92 | 29 | 1 | 30 | 0.97 | 35 | 3 | 38 | 0.92 | 35 | 2 | 37 | 0.95 |
| | I | 18 | 8 | 26 | 0.31 | 11 | 9 | 20 | 0.45 | 5 | 7 | 12 | 0.58 | 5 | 8 | 13 | 0.62 |
| | $\Sigma$ | 40 | 10 | 50 | | 40 | 10 | 50 | | 40 | 10 | 50 | | 40 | 10 | 50 | |
| | $P_A$ | 0.55 | 0.80 | | **0.60** | 0.73 | 0.90 | | **0.76** | 0.88 | 0.70 | | **0.84** | 0.88 | 0.80 | | **0.86** |

Misclassification of healthy trees for infested was more common in the first two periods (June and beginning of August); however, the remaining two periods (end of August and beginning of October) proved the possibility to apply vegetation indices calculated from low-cost UAV sensor for detection of infested trees with sufficient accuracy; details are tabulated in Table 3. For example, the omission error of the GI changed between the first and last sensing periods from 25% to 2% and the commission from

3% to 2% for healthy trees; for infested trees, the error of omission was 10% and of commission 53% to 10%. The accuracy of the MLC classification in distinguishing healthy trees from the infested ones was associated with the actual increase in the differences between the vegetation indices; the differences in variability did not play a major role (see Table A1 in Appendix B). The effect of the time point of image acquisition on the detection accuracy was more significant than the effect of the selection of a variable as vegetation indices are strongly correlated [42].

## 4. Discussion

The study results are consistent with those of other studies focused on bark beetle detection using UAVs and correspond with the findings of Abdullah et al. [14], who used a field spectrometer, and Minařík et al. [23] who used non-calibrated raw values. Our study complements those by Stoyanova et al. and Brovkina et al. [21,22] who investigated UAV-based detection capabilities on mosaics based on images acquired during a single mission. In the corresponding time of acquisition (with regard to a different latitude), we recorded a 78% overall accuracy. For example, Näsi et al. [4] achieved an overall accuracy of 76%. The overestimation of the number of infested trees was more common in our study while the opposite was true in the studies of Näsi et al. [4,19]. Due to their previous results documenting good distinctiveness between healthy and dead trees (the overall accuracy 90%), we focused our attention only on distinguishing between healthy and infested trees. Our findings also broadly correspond with studies using high-resolution multispectral satellite imagery, especially Landsat [12,43,44].

We assumed that compared to healthy trees, the infested ones have a higher reflectance in the Green and Red parts of the spectrum and lower in the NIR (Figure 2), which corresponds with the results of Näsi et al. and Abdullah et al. [4,14]. Therefore, the vegetation indices combining these bands were chosen for evaluation. The study results suggest that the Red band is crucial for detecting the bark beetle. The assumption about the benefit of NIR band acquired by a customized CIR camera wasn't proven (Hypothesis b), unlike in previous studies [4,19,21,23]. The non-calibrated CIR camera sensitive to the wavelength of approx. 760 nm used in our study seems to be insufficient for distinguishing between healthy trees and those infested by the bark beetle. Based on our results, we assume that it is necessary to use higher NIR wavelengths around 800 nm [14] for refining the results (with regard to accuracy and timeliness of detection). The Blue and Green bands recorded more or less a stable difference between healthy and infested trees in all four sensing periods. The vegetation indices derived from these bands also yielded only limited success in detection of the bark beetle infestation.

Radiometric normalization of the mosaic was conducted using the Flat Field Correction method due to the absence of targets for radiometric calibration during the field campaigns or an irradiance sensor on UAV. The main task of the correction was to normalize the differences between RGB and customized CIR cameras. The Flat Field Correction is based on dividing image pixels by the mean reflectance values calculated from the user-defined region of interest represented by a spectrally flat material. The boulders in the dry river basin were the only suitable type of this material in the study area. Its reflectance seems to be stable across the camera wavelengths. The result of this correction is the relative surface reflectance. Depending on the sensors' sensitivity and the reflectance of the reference area, this method may lead to relative surface reflectances with values higher than 1 (see NIR in Figure 2). Therefore, the comparison among the four periods (represented by vegetation indices) can be slightly affected by the choice of the ideal flat object in the orthomosaics. The possible inaccuracy of data normalization may be apparent in spectral profiles of the Period 4 (see Figure 2) where the value of NIR surface reflectance is slightly higher than for the remaining periods. On the other hand, it could be also caused by weather conditions because October is the moistest and rainiest month from sensing periods. Therefore, in reality, the general vitality of trees could be greater than in the remaining periods. The imagery normalization could potentially affect the period's statistical evaluation but could not influence the results of the MLC classification of the individual mosaics and the distinguishing between infested and healthy trees within one sensing period. Despite this crude

solution, the shapes of corrected spectral curves are very similar to those acquired by other authors using state-of-art multispectral [23] and hyperspectral [4,19] sensors. Another normalization option would be for example to use the methods based on chromatic coordinates [45] instead of the Flat Field Correction tool.

The results may also theoretically be affected by different lighting conditions during image acquisition (e.g., the amount of shadows). The flights in June and in October were performed in overcast conditions while the flights in August in sunny weather. For the study needs, however, it was important that the individual sets of images captured on the same date were acquired under constant conditions as we were comparing data with the same day of acquisition, not data between the periods. In the overcast periods, the cloud cover of the sky was consistent during each UAV mission and in sunny periods, almost no clouds that could cause problematic shadows were present in the sky. To further minimize the effects of differences in lighting conditions, all UAV flights were performed between 11:00–13:00 of the local time to minimize shadows. As individual flights take only 15 min, we were able to wait for the appropriate time of acquisition. The remaining shadows were masked and a mean value of vegetation index in 0.5 m buffers around the tree tops was calculated. Therefore, we assume that the weather had a minimum influence on study results. Other studies [4,19] solved this problem by using the average value of the six brightest pixels in a 1 m diameter window.

We did not consider the possible differences in the vegetation indices caused by differences in the tree sizes. However, Näsi et al. [19] who took that into consideration did not reveal any significant effect of such approach.

In our study, the object-based classification using the Maximum Likelihood classifier [46], representing one of the most widely used and approved classification approaches worldwide, achieved accurate classification results (Table 3). On the other hand, Näsi et al. [4] used the k-nearest neighbor classifier (k-NN) and recommended the use of the Random Forest algorithm; Näsi et al. [19] used the Support Vector Machine (SVM). The presented statistics is based on a representative sample of data and on a sophisticated and robust method, unlike previous similar studies such as the preliminary study by Minařík et al. [23]. The sample size in the study by Näsi et al. [4] was similar to ours. Due to a relatively small sample size of infested trees (compared to healthy ones) and the uniqueness of the study area due to its situation in the non-intervention zone, it is difficult to evaluate the universal applicability of this solution in everyday forestry practice. Same as Näsi et al. [4], we believe that we cannot simply proclaim our method to provide sufficient/insufficient accuracy of infested trees detection as that always depends predominantly on the particular needs of the forest managers.

Both statistical analysis and image classification show that for identification of infested trees, the use of a consumer grade RGB camera is sufficient (Hypothesis a). The negligible impact of the near-infrared band (in the study expressed as SR, GRVI, NDVI, and GNDVI indices) can be caused by a lower NIR spectral resolution (lower wavelength) of our customized sensor. However, we assume that using a professional UAV multispectral, e.g., Tetracam μ-MCA Snap 6 [23], or hyperspectral camera [4,19], could further increase the resulting significance.

There are further future possibilities to build on this study and to advance the knowledge of UAV-based detection of bark beetle infestation. It is necessary to confirm our results with regard to early detection of bark beetle infestation in other types of environments using different sensors. For precise detection, more information is needed about spectral characteristics of different stages of bark beetle infestation and their spatial-temporal changes, e.g., by using laboratory or field reflectance measurements [14], by UAV monitoring with high temporal resolution [4], or through time lapse cameras. Another possible challenge lies in the synergy of application of fine-scale UAV data with satellites (e.g., Landsat 8, Sentinel-2, or Planet) or airborne sensors, which can allow extrapolation of UAV results to larger areas.

Presented methodology describes a novel low-cost approach for fine-scale detection of bark beetle infestation using vegetation indices and the identification of different stages of the tree infestation. Nevertheless, the study results are applicable for detection of any biotic forest pest disturbances as

well as for other forestry applications where the spectral characteristics of the tree crown are crucial [4], such as tree species classification [22].

In our study area, the outbreak started on 1 June (till 10 June) and the new generation was born after 15 August, which corresponds with the dates of acquisition of UAV-borne images. At the second stage (1 August), the infestation was usually not easily recognizable from the ground; however, during the next stage (30 August), the spectral difference was significant, and the infestation was already recognizable from the ground. In the late stage (1 October), when the infested trees were almost dry, the spectral variation as well as color differences were obvious. Our results clearly show that the detection of infested trees was possible as soon as a few weeks after the outbreak. However, the period of early bark beetle attack detection is highly site- (lowland vs. mountains etc.) and condition-specific (temperature, humidity etc.). Using state-of-art multispectral and hyperspectral sensor [14] or full-wave LiDAR [30] could further improve the period between infestation and detectability and contribute to better distinguishing of bark beetle infestation at the earliest stage. Therefore, more studies have focused on other sites and the use of different sensors are needed to validate the findings of this study as well as to make the knowledge base wider. As the bark beetle infestation is a serious environmental and economic threat, we are planning further research in this field using professional multispectral sensors at various study areas.

## 5. Conclusions

Our study clarifies the potential of consumer-grade and customized sensors mounted on a UAV platform for the detection of bark beetle infestation in different stages throughout the season. Results indicate that even with low-cost UAV-based solutions, it is possible to precisely detect the infestation at the level of individual trees (Hypothesis a). We observed differences in the spectral response (based on vegetation indices) early after the bark beetle outbreak (in the so-called green attack stage), i.e., in the period when the infested tree cannot be easily recognized from the ground. The Greenness Index yielded the most promising results. Conversely, a performance of the indices based on the near-infrared band was lower, therefore we conclude that RGB bands (or Red band) play a more important role in the detection (Hypothesis b). In addition, results confirm our assumptions that with increasing time after infection, it is easier to distinguish between healthy and infested trees. For the management of roughly hundreds of hectares, such an early detection is crucial; however, if thousands of hectares need to be managed, the few weeks of early detection may prove insignificant and prediction might be more important. Nevertheless, further research assessing the spectral characteristics of infested trees after an even shorter time (very early stage) from the outbreak using state-of-art sensors is still necessary.

**Author Contributions:** All authors contributed in a substantial way to the manuscript. T.K. and J.K. conceived, designed and performed the experiment and wrote the manuscript. K.H. performed the statistical evaluation. P.J. and B.V. acquired input data. P.S. invented, supervised and discussed the research. All authors read and approved the submitted manuscript.

**Funding:** The study was supported by the Technology Agency of the Czech Republic under the grant No. TJ01000428, grant "EXTEMIT - K", No. CZ.02.1.01/0.0/0.0/15_003/0000433 financed by OP RDE and the Czech University of Life Sciences Prague under grant No. GIGA 20184206.

**Acknowledgments:** We acknowledge three anonymous referees for their constructive comments. Also, many thanks for the helpful comments to our colleagues from the Department of Applied Geoinformatics and Spatial Planning at the Czech University of Life Sciences Prague (CULS). We would also like to thank Michal Fogl for help with data processing and image creation; we also would like to thank Katka Gdulová for map preparation. Finally, we would like to thank many times our colleague Václav Jansa from The Krkonose Mountains National Park Administration.

**Conflicts of Interest:** The authors declare no conflict of interest.

## Appendix A

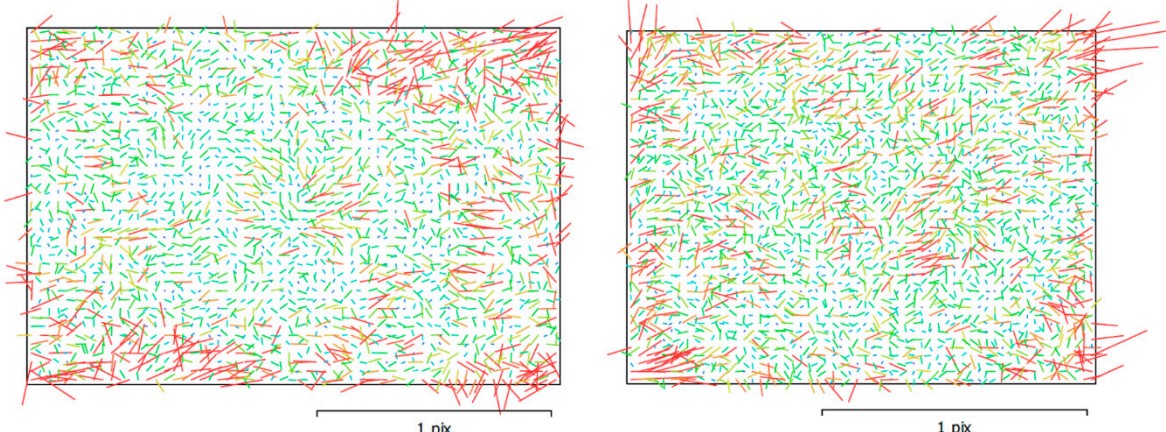

**Figure A1.** Image residuals for Sony A7 (left) and Lumix TZ7 (right) side after image-matching processing.

## Appendix B

**Table A1.** Comparison of the variability of the training data reflectance values. H: represents training H-sample and I: training I-sample. First statistical value shows their mean, the second standard deviation for every used vegetation index and sensing period.

|  | Period 1 | Period 2 | Period 3 | Period 4 |
|---|---|---|---|---|
| **SR** | **H:** 2.090 ± 0.299 | **H:** 2.065 ± 0.328 | **H:** 1.817 ± 0.320 | **H:** 2.226 ± 0.439 |
|  | **I:** 1.901 ± 0.231 | **I:** 1.653 ± 0.215 | **I:** 1.364 ± 0.146 | **I:** 1.530 ± 0.283 |
| **GI** | **H:** 1.132 ± 0.043 | **H:** 1.114 ± 0.041 | **H:** 1.096 ± 0.036 | **H:** 1.138 ± 0.059 |
|  | **I:** 1.105 ± 0.043 | **I:** 1.073 ± 0.044 | **I:** 0.995 ± 0.036 | **I:** 0.929 ± 0.051 |
| **GRVI** | **H:** 1.844 ± 0.223 | **H:** 1.849 ± 0.240 | **H:** 1.652 ± 0.249 | **H:** 1.945 ± 0.301 |
|  | **I:** 1.717 ± 0.160 | **I:** 1.535 ± 0.148 | **I:** 1.368 ± 0.105 | **I:** 1.638 ± 0.220 |
| **NDVI** | **H:** 0.347 ± 0.064 | **H:** 0.340 ± 0.068 | **H:** 0.281 ± 0.081 | **H:** 0.369 ± 0.085 |
|  | **I:** 0.306 ± 0.062 | **I:** 0.241 ± 0.064 | **I:** 0.151 ± 0.050 | **I:** 0.201 ± 0.078 |
| **GNDVI** | **H:** 0.292 ± 0.057 | **H:** 0.293 ± 0.058 | **H:** 0.239 ± 0.071 | **H:** 0.314 ± 0.070 |
|  | **I:** 0.261 ± 0.048 | **I:** 0.208 ± 0.047 | **I:** 0.154 ± 0.037 | **I:** 0.237 ± 0.060 |

## Appendix C

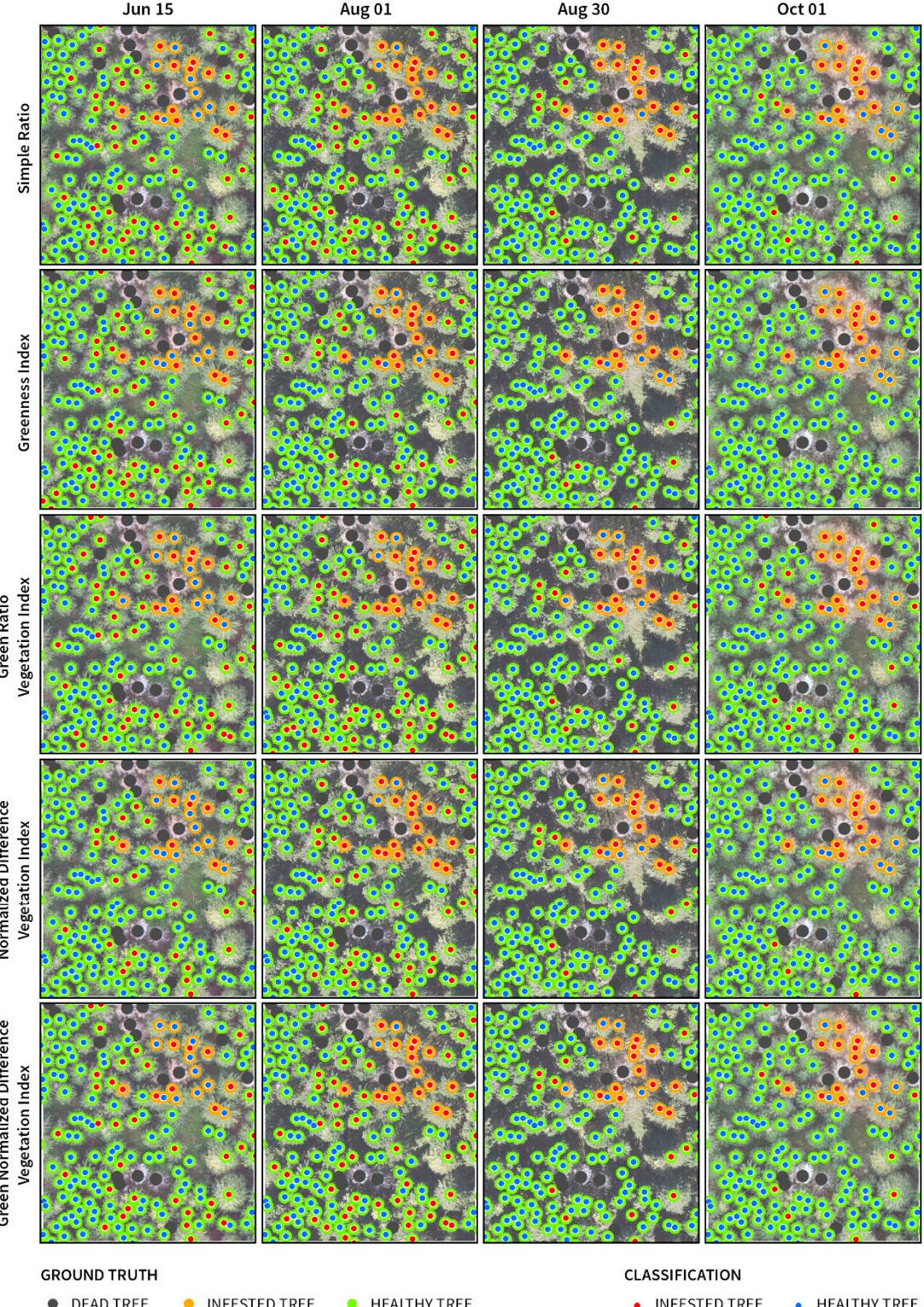

**Figure A2.** Illustration of the performance of the Maximum Likelihood Classifier (MLC) classification based on the different vegetation indices in four unmanned aerial vehicle (UAV) sensing periods. The red dots in the figure represent infested trees in different stages of the bark beetle attack, blue dots represent healthy trees. The trees without dots are dead trees infested in previous years. The illustration represents only a small subset of the study area.

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
