# Peer review of "The Use of UAV Mounted Sensors for Precise Detection of Bark Beetle Infestation"

_remotesensing, doi:10.3390/rs11131561_

Round 1
Reviewer 1 Report
The subject of the manuscript is interesting, although UAV use in monitoring the bark beetle has little practical application due to the possibility of analyzing only a small area.
The paper is well prepared, but in some places you can find trivial statements, as well as some important information missing:
1. In lines 88-89 the authors write: "near-infrared band plays an important role in the detection of spectral changes" - this statement is trivial and obvious.
2. In lines 136-137 the authors write: "Shadows were masked using the near-infrared band and CHM model during the thresholding process" How was the near-infrared band used?
3. lack of information on variation of the reflectance values for training samples - the maximum likelihood classification method tends to overstate classes with high variability. This information would be helpful in analyzing / discussing the accuracy results of the classification of individual vegetation indicators. Probably misclassification in the first two periods results from significant differences in the variability of training samples for healthy and infected trees.
4. no legend in Figures 4 and 4b - it is not clear what the dots mean.
5. The lack of consistency in naming classes - in table 3 the authors used other class names than in the earlier part of the manuscript - C0 and C1 instead of G and H.
Reviewer 2 Report
The authors present a method to detect bark beetle (Ips typographus) infested trees in the green attack stage with UAV data captured with two different cameras: (1) a standard RGB camera and (2) a camera modified to capture NIR images. UAV data were collected four times during an infestation to study how early the infestation could be detected in the images. Statistical tests were performed to study the separability between healthy and infested trees, and a maximum likelihood classification was performed to classify trees into healthy and infested with an accuracy of 78% at the early stage and 96% at the late stage of the infestation.
The study is interesting with UAV data captured at four different times with a standard RGB camera but some methods are poorly described and needs to be clarified. The article would also need language improvement and changed structure. Now the discussion is part of the results and not a section on its own (Section 3.4).
Major comments:
Radiometric correction is crucial when comparing images from different flights especially with longer time periods between the flights. The authors need to provide more details about the radiometric correction. How could you use a flat field correction to get reflectance? Flat field correction is used to compensate for differences in pixel sensitivities etc. in the sensor. To get reflectance, areas with known reflectance in the images must be known (unless the incoming radiance is measured).
Related to the radiometric correction are the weather conditions during the flights. What were the conditions, sunny, cloudy? The authors need to describe the weather conditions when flying. The images in the appendix indicate that the August images were captured in sunny conditions (strong shadows) while the other images were captured in cloudy conditions (no shadows). Varying light conditions will have a strong influence on the images (and in turn the spectral signatures) and needs to be handled.
Also the differences in sun angles are different in mid-June and mid-October. That will have an influence on the images, especially in hilly areas, that at least needs to be discussed.
The comments related to radiometry above could potentially have a major influence on the spectral signatures and the statistical analyses of separability.
Another important part that is missing is how the spectral signatures were created. Are the red, green, blue bands from the RGB camera and the NIR band from the modified camera? That would mean the signatures are created from images from different flights even if they are from the same day. That part must be clarified. How was the modified camera modified? By the authors or by the manufacturer? What wavelength band (roughly) did the NIR camera capture?
One important note about the bark beetle infestation is when the outbreak started. When do bark beetles usually swarm in the area? If they swarm e.g. in early May the first images are already some weeks after the swarming and not from before the outbreak (attack). It does not seem likely that the bark beetles swarm after June 15 in the area? This would mean the infestation is in a later stage then what the authors suggest and that the actual detection of the infestation is in a rather late stage (close to red attack rather than green attack). It is also interesting to know if there are typically two generations of bark beetles in one year.
Reflectance in the red, green and blue wavelength bands are lower for the healthy than the infested trees at all four flights (Figure 3). What can be causing this? Did the infestation start well before the first flight so there was already an influence from the infestation? But it seems that mainly the red band is influenced by the infestations (difference between infested and healthy is increasing in the red band but more or less stable for the blue and red for the four flights). The consistently lower reflectance for healthy trees compared to infested should be discussed (for RGB).
One interesting analysis that the authors could do would be to check how the Red Chromatic Coordinate (RCC) change between the four flights. RCC (and GCC; Green Chromatic Coordinate) have been used e.g. for phenocams and phenology (See e.g. Richardson, A.D., Hufkens, K., Milliman, T., Aubrecht, D.M., Chen, M., Gray, J.M., Johnston, M.R., Keenan, T.F., Klosterman, S.T., Kosmala, M., Melaas, E.K., Friedl, M.A., & Frolking, S. (2018). Tracking vegetation phenology across diverse North American biomes using PhenoCam imagery. Scientific Data, 5, 180028). In this study, using RCC, would help to adjust for limitations in radiometric corrections and since most of the changes seem to happen in the red wavelength bands it is likely that RCC would be useful for the study to limit the influence of varying light conditions. It could be a more robust (concerning radiometry) compared to the methods applied by the authors since RCC compensates for differences in incoming light. Maybe RCC alone would give a good result?
A Maximum likelihood classification with one band only is used to classify the trees into healthy and infested. Why did the authors use a maximum likelihood classification when only one band was used for the classification? Maximum likelihood is a method for classifying data with several wavelength bands; with one band it will more or less be like using a threshold value for the classification.
The authors also suggest that a threshold method could be applied for the classification since that would have the advantage that training data are not required (P10, L296). How would that be performed, i.e. how can a threshold be decided without any training data available for healthy and infested trees? (The cited reference is not in English).
The authors states that they created a digital terrain model (DTM) and from this a canopy height model (CHM). How was this done? In forested areas with that little ground visible in the images it is not really possible to create an accurate DTM.
It is stated that the individual tree tops “were automatically detected using ArcGIS” (P4, L138-139). How was the detection performed? It must be explained.
Minor comments:
P1, L19: one of endangered ecosystems. Please reformulate.
P1, L26: UAV-borne imagery was matched. Please reformulate.
P1, L36-37: Keywords are not relevant. Please change.
P2, L64: Invasive approaches?
P2, L72: “…those were applied...”. Change to “…those have been…”
P2, L74: Only spell out UAV first time it is mentioned (check entire manuscript).
P3, L96-97: Latin names within parentheses.
P3, 104: Overview map is a bit hard to read. Maybe make the dark colors brighter?
P3, L112: UAV equipped with rather than “mounted with”.
P3, L117: What was the overlap? “regular overlap” is not very informative.
P3, L122: UAV-borne images. Please reformulate.
P4, L125: “…were built in Ground…” Use “with a” rather than “in”.
P10, L262: “…studies focused…”. Change to “have focused”.
P10, L262: “…using a hyperspectral…”. Remove “a”.
P10, L284: “…may be also…” Change to “may also be”.
P11, L313: “…environment using…”. Change to “environments”.
P12: What is the figure in the Appendix showing? Add at least a caption.
There are several other minor language corrections that need to be performed.
Reviewer 3 Report
General comments
The manuscript assesses the potential of low-cost UAV-mounted sensors to detect bark beetle infestation. It also analyzes the potential for an early detection, and test which remote sensing derived variables would be the most important for this application. Overall, I like it, I find the manuscript well-written, concise, easy to read and interesting. I have a list of suggestions (below) and a few major concerns:
1- In Introduction, it is missing a better description of what past studies have done on the subject and what is the novelty of the study taking those in consideration. The manuscript already has some text for this in the discussion section that could be used.
2- In results, the first section 3.1 is too subjective compared to the rest of the analyses, and I am not sure the findings are correct. I suggest the authors to check the results and include a more quantitative approach, describing the spectrum values (i.e. mean, sd, confidence interval, etc.) and assessing whether they are different between infested and healthy trees.
3- Related to the discussion of point (2), I expect that the red band would be the one that most distinguish the classes, and that should help explain why the vegetation indices that used the red band had better results for this application – consider adding a sentence to the Conclusion. If that is the case, you should add an explanation on why the red band was important, and also why the NIR band was not important (since this was one of your hypothesis). Some ideas to work on and find references, for example (1) for red, changes in chlorophyll content, and (2) for NIR, if it is observed in the field that the leaves/needles don’t fall right after with the infestation, maybe it was not supposed to have a major NIR change within the time period at all.
Introduction
L45-46, The way you cite the 4 years is vague, the time period (from year X to Y) is missing. You could also mention this sentence as an example of factors that can increase the probability of the beetle’s attack.
L74, You have cited the studies [4,21–24] that used UAV-based data to detect beetle infestation. I see that you commented in L76-77 how these studies are still limited. More information on what these studies have done are missing here, e.g. what kind of sensors do they use, what accuracy they achieve, their limitations. For instance, by reading this paragraph and reading your objective, it is not clear that what you are doing is new regarding to what they did.
L84-89, I imagine that you would like to test if the NIR band is important for the detection because NIR signal is often related to forest structure. However, you did not refer to the NIR signal in the text beforehand. You should comment on previous uses of NIR for this application.
L84-89, since you state two hypothesis, check later on in the text if you reject them or not.
Methods
Figure 1, In the label indicate what is this image, is it a true color composite obtained from the UAV, which date.
L108-116, include the hour of day of acquisitions as this would influence the shadow distribution in the imagery. If the data were acquired at different hours, justify how this would not impact the analysis
Figure 2, I suggest removing this figure or moving it to the supplementary material – I don’t find it very interesting for the analysis
L137-139, Missing more info on tree top detection, which method does ArcGIS uses? Which band(s) were used in this process?
L140, based on what do you say they prefer trees older than 60 years and how do you reach the 15 m threshold for this age? Should add some reference here for the age-preference, say something about the height distribution in the area, and define the age/height distribution relationship
L143-144, remove the parenthesis, make of this a new sentence, i.e. “This was based […]”
Figure 3, A few things here. 1) Add the legend for the three lines in the figure for easier visualization; 2) Missing y-axis scale; 3) Consider adding the standard deviation or confidence interval for each value because only the mean is not enough to visually assess whether the classes are distinguishable; 4) how many trees each line represents (n = ?); 5) the color of points in the upper-row is misleading, maybe it should match the lines color (?); 6) separate the x-axis “line” between plots, because it can be a bit confusing at a first look
L160-167, do all trees went through the beetle attack at the same time?
Results and discussion
L193-195, This sentence is possibly wrong. In Figure 3 in the visible spectrum the infested and healthy have indeed some difference in the mean values, but they are very similar in the NIR band. Another thing, to assert whether these differences are distinguishable you should apply some statistical test or at least describe some statistics such as mean and confidence interval.
L193-195, It is also not clear to me if you are comparing in Figure 3 the spectrum from only three samples (one each class) or from all the trees. I suppose the analysis should encompass all the trees.
L195-196, This sentence “The spectral response […]” is missing something
L197-198, exactly how do this complement the other studies?
L201, “The response of both infected and healthy trees changed over time”. Justify why.
L201-203, “shapes of the spectral curves were relatively similar due to the similar spectral response”. This is kind of redundant, consider re-writing and be precise.
Figure 5, some points: 1) indicate y-axis label; 2) use points instead of commas in y-axis decimals; 3) the figure must be self-explanatory, so you should consider distinguishing the infested and healthy trees in the figure, e.g. with a different color or indicator
L248-249, You could include here in the text a rough estimate of omission and commission errors for the first and last periods – maybe for the best variable. I understand that the results are all in Table 3 but there are lots of numbers in there, and it may be confusing for the reader.
L262-265, Consider moving this to introduction, and then you just compare your findings to theirs here.
L277-289, It is good that you discussed the radiometric normalization and shadows here, it was indeed needed. Just wondering, couldn’t you try an inter-image normalization approach between the four images using (more or less) spectrally stable targets?
L318, Data from Planet satellite constellation with 3-m spatial resolution and frequent acquisition could be even better than Landsat and Sentinel-2. Its spatial resolution is closer to the crown size, and the frequent image acquisition should more closely observe the changes in the infected trees signal
L325-331, it is a writing style matter, but sometimes this kind of paragraph fits as the last paragraph of introduction to highlight the novelty of the work
L332, so is the obtained early detection fast enough for forest managers?
Appendix A, Some points: 1) it has some strange borders; 2) it is missing a caption describing it; 3) it is hard to readily identify which trees are correctly identified as infested or healthy because there is no indicator of which trees were identified as infested in the field. Could you add this in the figure? An idea could be to differentiate the correct/incorrect with solid/empty circles.
Round 2
Reviewer 2 Report
Dear authors,
You have done a very good job with the updated manuscript and responses to comments.
Please change "draught" on P2, L46 to "drought".
Reviewer 3 Report
The manuscript has been significantly improved with the revisions. I only have two minor comments to be considered:
1) L246-249, these two last sentences look like discussion
2) In Appendix C, Figure C1, I can only see the Ground Truth classes 'Dead Tree' and 'Healthy Tree', I cannot see the 'Infested tree' class in orange in the figures. Consider checking if it is correctly plotted, or maybe try changing the colour for better contrast.
